# Post-COVID-19 Spondylodiscitis: A Case Study and Review of the Literature

**DOI:** 10.3390/medicina59030616

**Published:** 2023-03-20

**Authors:** George-Cosmin Popovici, Costinela-Valerica Georgescu, Anca-Adriana Arbune, Mihaela-Camelia Vasile, Ionut Olteanu, Manuela Arbune

**Affiliations:** 1School for Doctoral Studies in Biomedical Sciences “Dunarea de Jos” University from Galati, 800008 Galati, Romania; 2Pneumophtiziology Hospital Galati, 800189 Galati, Romania; 3Pharmaceutical Sciences Department “Dunarea de Jos” University from Galati, 800008 Galati, Romania; 4Gynecology and Obstetrics Clinic Hospital Galati, 544886 Galati, Romania; 5Neurology Department Clinic Institute Fundeni Bucharest, 022328 Bucharest, Romania; 6Clinic Hospital for Infectious Diseases Galati, 800179 Galati, Romania; 7Emergency University Clinic Hospital Bucharest, 050474 Bucharest, Romania; 8Medical Clinic Department “Dunarea de Jos” University from Galati, 800008 Galati, Romania

**Keywords:** SARS-COV-2, COVID-19, spondylodiscitis, spinal epidural abscess, MRSA

## Abstract

COVID-19 is currently a major health problem, leading to respiratory, cardiovascular and neurological complications, with additional morbidity and mortality. Spinal infections are rare, representing around 1% of all bone infections and comprising less than 2 per 10,000 of all hospitalizations in tertiary care centers. Spondylodiscitis is a complex disease, with challenging diagnosis and management. We report the case of a 45-year-old man, non-smoker hospitalized for severe COVID-19 disease with respiratory failure. Post-COVID-19, in the 8th week after discharge, he was diagnosed by magnetic resonance imaging with spondylodiscitis, but etiology was not confirmed by microbiological investigations. Antibiotics were used, considering the identification of MRSA from cultures of pleural fluid and nasal swab, but surgical intervention was not provided. Clinic, biologic and imagistic were improved, but rehabilitation and long term follow up are necessary. We concluded that spondylodiscitis with spinal abscess is a rare but severe complication post-COVID-19 disease, due to dysbalanced immune response related to the respiratory viral infection, endothelial lesions, hypercoagulation and bacterial superinfection.

## 1. Introduction

The coronavirus disease 2019 (COVID-19) emerged as a primarily respiratory tract infection, but soon, the clinical and pathological data evidenced the systemic characteristics and extrapulmonary increasingly expression [1].

The etiological agent is SARS-COV-2, which can damage the endothelial cells, increasing the risk of vascular coagulation, even in moderate category of disease, and also paucisymptomatic or asymptomatic cases [1,2].

In a large series of autopsy, all cases presented variable degree of endothelial damage and 87% of cases evidenced arteriolar thrombosis [2,3,4].

Essentially, COVID-19 consist of a complex clinical disorder including disseminated micro-embolisms, bleeding diathesis, diffuse vasculitis, autoimmune aggression and decreased antibacterial defense.

Lymphocytes and monocytes are often decreased, impairing the immune response to other bacterial or fungal infectious agents [5,6].

The treatment of COVID-19 has evolved according to the progress of knowledge on the pathogenesis of the disease and the results of clinical studies envolving different pharmaceutical molecules, the following categories of drugs such as: antiviral, anti-inflammatory and immuno-modulatory, anticoagulants, antibiotics and other infectious medications, supporting vital functions, symptomatic medication. The recommendations for each therapeutic intervention depend on the severity category of the disease and the patient’s risk factors [1].

The classification of symptomatic forms of COVID-19 includes mild (40%), moderate (40%), severe (15%) and critical (5%) forms. Severe forms require oxygen support, and critical ones evolve with complications, such as respiratory failure, sepsis, thromboembolism and multi-organ failure [1,7].

The post-COVID-19 syndrome, also called “Long COVID-19”, includes a series of persistent conditions in the following 12 weeks after an episode of COVID-19, among which: fatigue, dyspnea, cough, sleep and concentration disorders, anxiety or depression [8]. Recognizing and evidences of the emerging post-COVID-19 syndrome requires a larger number of data provided from case reports and clinical trials [1]

Spondylodiscitis associated to COVID-19 are new perspectives of post-COVID-19, rarely reported in the medical literature [9].

Spondylodiscitis (SD) is an infectious inflammation involving the vertebrae, vertebral discs and adjacent structures, with pyogenic, granulomatous (tuberculous, brucellas, fungal) or parasitic features. Yearly incidence of SD ranges between 0.2 and 2.4 new cases per 100,000 people, but is three times more frequent in men than women [9,10]. Practically, SD is rare, although the diagnostic could be underestimated, mostly in poor regions, with poor hygiene and difficult access to health care facilities

The SD can occur at any age, while is most often found in the sixth decade of life [11].

Moreover, the risk factors of SD are old age, diabetes mellitus, steroid therapy, rheumatic diseases, spinal surgery, kidney failure, liver cirrhosis, cancers, intravenous drugs use, intravascular devices, HIV or various infections of skin and soft tissues, mouth mucosa, genitourinary or respiratory tract and the source of infection is often undetected at diagnostic time [12,13]. The etiology could be bacterial or non-bacterial. The largest proportion of spondylodiscitis cases is caused by Staphylococcus aureus (20–40%), following *Enterobacteriae* spp. (7–33%), *Streptococci* spp., *Enterococci* spp. (5–20%), and anaerobic germs (less than 4%) [12,14,15].

Before the antibiotics use, spondylodiscitis thought to have led to mortality of 25–71% of cases, but nowadays is decreasing to 2–12% [12,16]. The pathogenic mechanisms are mainly blood-borne infections, either in spontaneous or post-operative SD [17]. The most common location of spondylodiscitis is the lumbar (60%), the breast (30%) and the cervical spine (10%) [16,18].

The diagnostic of SD is based on clinical, laboratory and imaging criteria.

Early stages of SD are characterized by nonspecific symptoms such as mild fever, general malaise, weakness, and weight loss. The main complaint is the back pain, that is intensifying at night, although the pain could miss in 15% of cases [19]. The diagnostic of SD is often delayed, due to the low specificity of clinical presentation [17,20]. The occurrence of complications, as sepsis, multiorgan failure or neurological symptoms could be related to para or intraspinal abscess or destruction of vertebrae bone, rising the mortality up to 17% [20].

High titers of C-reactive protein (CRP) and increased white blood cell count are usually markers of infections, although their significance is limited for the detection of SD [20,21].

The accurate diagnosis requires histological analysis or direct detection of aetiologic germ, from the blood cultures or cultures of biopsy samples [11]. Blood culture is the easiest, cheapest, and most effective method for confirming the etiology, if is obtained before the antibiotic treatment.

X-ray images of the spinal segments have occasionally associated osteolysis and shadowing of the paravertebral soft tissue, suggesting of a spinal abscess. Magnetic resonance imaging (MRI) is more sensitive and specific for the diagnosis of SD and therefore is the method of choice [11]. Paravertebral abscess could be a complication of spondylodiscitis, mostly appeared in hospitalized patients with an estimated incidence from 1.2 to 3 per 10,000 patients [22].

A half of cases has normal white blood cell and no fever, while bacteremia is detected in 60% and Staphylococcus aureus is responsible for 70% of cases. Vakilit [23].

The antibiotic conservative treatment is the standard of care for SD, but the failure rate range between 25% to 56%. There are no guidelines for the duration of therapy, but patients typically require 4–8 weeks of therapy [11,16,17]. The surgical treatment is recommended in patients with cauda syndrome, extensive bone destruction or antibiotic treatment failure. Minimally invasive surgical procedure is more and more advised [11].

The aim of our study is to analyze the case of a patient hospitalized with severe COVID-19 infection and spondylodiscitis with paravertebral abscess diagnosed during 12 weeks after the acute coronaviral infection. The link between COVID-19 and spondylodiscitis is explored considering the peculiarities of the case, the treatment with high doses of cortico-therapy, antiviral and immunomodulator treatment, and the review of other reported cases with the same comorbidities.

## 2. Case Report

We present the case of a 45-year-old Caucasian male, non-smoker, occasional consumer of alcohol, with professional toxic respiratory exposure and a personal medical history of appendicectomy, duodenal ulcer and lumbar disc herniation. He was admitted to a hospital from Romania in April 2021 for fever (maximum 39.7 °C), abdominal pain and diarrhoea, with onset of 10 days. Furthermore, he found at home a positive result of self-test COVID-19 antigen and he notified worsen general condition, fatigue, chest constriction and dry cough after a 5-day course of self-treatment with Clarithromycin, acetaminophen and acetylsalicylic.

The clinical examination revealed altered general condition, body mass index = 27.3 kg/m^2^ Glasgow coma score 15, pale skin, high fever of 40.1 °C, RR = 24/min, SpO_2_ = 92% with supplemental oxygen flow of 5 L/min by nasal cannula, rhythmic heart sounds, HR = 87 b/min, no lower limb oedema. The COVID-19 diagnostic was confirmed by a positive test RT-PCR-SARS-Cov-2. The thoracic Rx examination evidenced suggesting images of COVID-19 pneumonia (Figure A1).

The first line antiviral treatment was Favipiravir, but it was replaced in the second day with Remdesivir for a course of 6 days, due to the worsening of general condition and decreasing of oxygen saturation to 84%. Supplement of oxygen flow to 20 L/min (by mask) was necessary to increase the SpO2 to 96%. Although the lab tests for interleukin 6 and D-Dimers were not available, we considered the clinical course was correlated with high values of neutrophile/lymphocyte ratio and other inflammatory markers as predictors for “cytokine storm” and three doses of Tocilizumab 800 mg every 12 h were managed. According to the treatment local protocol for severe COVID-19 pneumonia, Ceftriaxone for 10 days was the antibiotic of choice. Pathogenic treatment with Dexamethasone for inflammation and Enoxaparin for hypercoagulation were used during the hospitalization. Other drugs were provided as antitussives, antipyretics, antacids and hepatoprotectives. The clinical state was progressively improved, with oxygen withdrawal and normalization of inflammatory syndrome (Table A1). The patient was discharged with recommendation of Apixaban 2.5 mg bid for anticoagulation.

Shortly after discharge, he complains lumbar pain with irradiation in the pelvic limbs and marked functional impotence that required medical investigations. The spinal magnetic resonance imaging (RMI) examination with contrast described a lesion process centered at the level of the intervertebral disc T11–T12 highly suggestive of a spondylodiscitis with paravertebral abscess on the left side (Figure 1, Figure 2a,b and Figure 3a,b).

A computer tomography (CT) examination of the chest and abdomen evidenced massive pleural effusion with passive atelectasis of the left lung parenchyma, right pleural effusion in small amounts, mediastinal lymphadenopathy, and right lung images corresponding to post-COVID-19 residual inflammation.

A catheter was applied for the left thoracic drainage and 2700 mL of pleural fluid was evacuated. The bacteriological examination was negative for tuberculosis, but a positive culture of Methicillin-resistant Staphylococcus aureus (MRSA) was reported, concomitant with MRSA nasal colonization. The bronchoscopy ruled out the proliferative appearance or suspected dysplasia and concluded the aspect of diffuse bronchitis. Bronchoalveolar lavage (BAL) was performed in the territory of left inferior lobe and right superior lobe and consecutive cytology examination support relatively rare lymphocytes and very rare mesothelial cells. Adenosine deaminase in pleural fluid has normal values, PCR-TB and QuantiFERON-TB test were negative.

A percutaneous CT guided vertebral biopsy in the T11/T12 disc was performed, with microscopic description of cartilaginous disc tissue, small fragments of bone tissue, proximal to a fibrin-haematic clot and a small epidermal cyst. Microbiological investigations of the biopsy sample were not available. There was not available another vertebral biopsy.

Other investigations were negative, including tumoral markers (alfa-fetoprotein, CEA, CA19.9, CA15.3), autoimmune markers (antiphospholipid, anticardiolipin, lupus anticoagulant antibodies), viral markers (HIV, hepatitis B or C).

During post-COVID-19 multidisciplinary investigations for the back pain, the patient experienced 2 short courses of antibiotic associations: 3 days of Vancomycin, Meropenem and Levofloxacin, 3 days of Ceftriaxone, Vancomycin and Ciprofloxacin. However, the pain score and the inflammatory markers has been increasing in the following 8 weeks after COVID-19 hospitalization.

The data assessment supported the diagnostic of T11/12 SD and left paravertebral abscess, although the aetiology has not been identified. The therapeutical decision considered the context of positive culture of pleural fluid and nasal swab and the statistical probability of staphylococcal aetiology in at least 50% of SD cases. The recent history of severe COVID-19, previous treatment with Ceftriaxone and the characteristic of methicillin-resistance have justified Vancomycin and Rifampicin for the first line of antibiotic treatment [17]. While the clinical condition and biology improved in 8 weeks of antibiotic, the follow-up after 6 weeks evidenced relapse by increasing of inflammatory markers and pain score (Table A2). Meanwhile, the imagistic recovery was not significant. The neurosurgical drainage was not regard as opportune, due to the small size of the paravertebral collection and absence of neurological deficit, and continuing conservative treatment was recommended. The TB-cultures kept on negative in 8 weeks. Echo-cardiography and cardiologic examination excluded concomitant endocarditis. Although the patient continued to be afebrile there were prevailed 4 sets of haemocultures that were negative. We decided a second line of treatment for MRSA with Linezolid and TMP-SMX for other 8 weeks [12]. The evolution was favourable, with biological normalization, but the pain is incompletely retrieved and will be managed by rehabilitation programmes [Figure A2].

## 3. Discussion

Spondylodiscitis and/or epidural abscess are rare diagnostics in general population and the relationship with COVID-19 is not clarified. Nevertheless, it is expected that these conditions are underdiagnosed, in COVID-19 patients with coma or severely compromised, when other vital clinical interventions are prioritized.

### 3.1. Systematic Review of Medical Databases

Systematic research by key words “COVID-19 and spondylodiscitis/epidural abscess” identified in December 2022 only 4 publications with case reports or cases-series, from Pubmed, Google Scholar and Web of Science Core Collection databases, in the fields of medicine general internal (2), clinical neurology (1), orthopaedics (1) and surgery (2) (Figure A3). There were not found previous systematic reviews or metha-analysis. According to PRISMA diagram, the cases with fungal aethiology and the repots with bacterial SD and/or epidural abscess diagnosed over 12 weeks post-COVID-19 were excluded. [24,25,26,27,28,29,30].

The age of patients ranged between 24 and 78 years old (average 55,27), were predominantly males (7/11) and had associated comorbidities, mainly blood hypertension (5/11). It could be supposed that vascular microbial penetration during bacteraemia is facilitated by the damaged vascular endothelium, related to COVID-19 and arterial hypertension.

The potential iatrogenic immunosuppression induced by Tocilizumab (antagonist of the interleukin-6 receptor), hydroxychloroquine or corticosteroids, after therapeutical used for severe COVID-19, was notified in a third of cases (4/11). Although immunosuppression could be a favourable condition for the development of SD, the severity of COVID-19 appears not critical. The distribution of spine lesions (SD and/or epidural abscess) by the anatomic site evidenced by MRI have uniform involved the cervical, thoracic, and lumbar region, usually corresponding to multiple vertebral units. Back pain was the constant symptom of SD, while the rate of fever was 27.7%. Unknown microbial aetiology was assumed in three cases. Tuberculosis was found 12 weeks post-COVID by histopathology examination, in a previously heathy young woman. The main aetiology was MSSA, in two thirds of cases (7/11), documented by culture from the aspirate or biopsy of the lesion, while haemocultures were positive simply in three of these cases. The empiric antibiotic treatment was specified in 3 from 10 nontuberculous infections, and two of them ended as unidentified aetiology. Antibiotic according to the susceptibility testing was chosen in six patients, and replaced the first line in one other case, but all cases experienced surgical specific interventions. Excepting one death by multisystem organs failure and one case of deterioration by pulmonary thrombosis, the other patients noticed remission of infection and clinical improvement (Table 1).

### 3.2. Particularities of the Reported Case

Referring to our case presentation, we have identified several peculiarities and open questions. He was a younger male than the average age of the previous reported cases. Although his past medical history was not relevant for risk factors of severe COVID-19, he experienced severe pneumonia, requiring complex treatment, including corticosteroid, immunomodulator with Tocilizumab, and oxygen flow supplementation. Immunosuppression was consecutive to severe viral infection and iatrogenic interventions during hospitalization, but professional toxic respiratory exposure should also contribute to severe lung expression of COVID-19.

Immunosuppression was consecutive of viral infection and iatrogenic procedures during hospitalization, but the serious lung expression of COVID-19 might also be influenced by the professional respiratory toxics. Moreover, we noted a history of lumbar disc and herniation, that could point a predisposing condition for discal disfunctions. MRI decided the diagnostic of thoracic SD and paravertebral abscess, but the aetiology was not identified neither in haemoculture, nor in the spine biopsy lesions. The probability of staphylococcal spinal infection was considered, due to the bacterial evidence in positive cultures of pleural fluid and nasal swab, and according to the statistical data. Different from MSSA of reported cases, MRSA was isolated in our patient and probably involved in SD. The source of MRSA is supposed to be related to hospital, according to the case definition of health care associated infections [31].

Persistent or recurrent staphylococcal bacteriaemia related to SD in the COVID-19 pandemic has been reported in other cases of young healthy patients and unapparent infectious foci, speculated to be accounted to the immune disfunction, with secondary (reactive) hemophagocytic lymphohistiocytosis and to the diffuse vascular damages [32,33,34,35]. The first line of anti-staphylococcal antibiotics aimed highly antibiotic resistant MRSA but resulted in incompletely recovery after six weeks. Additional course of second line antibiotics was necessary. Our patient has persistent back pains, but no neurological deficit, and the surgery was not accepted as a treatment option, in contrast to the reported cases discussed above. Long-term impact on quality of life will be follow up [36].

### 3.3. Hypothesis of the Linkage between COVID-19 and Spondylodiscitis

The linkage between COVID-19 and spondylodiscitis is not clear, although pathogenic mechanisms are hypothesized. Spontaneous bacteraemia disseminated bacterial pneumonia or other bacteraemic hospital associated infections during COVID-19 could be the source of spondylodiscitis. Immunosuppression induced by the viral infection, the corticotherapy or immunomodulator agents are generally increasing the susceptibility for infections. Particularly for severe COVID-19 pneumonia, the respiratory disfunction associate increased intrathoracic pressure, microtrauma of the small vessels and haematomas of ligamental or muscular tissues, that could be favoured by the anticoagulant treatment [37]. The epidural vascular plexus is communicating with thoraco-abdominal venous vascularization and epidural or vertebral discs haematoma could be ruptured during the intense cough, with consecutive adjacent necrosis [37]. In vivo experimental studies on porcine models regarding the pathophysiology of intervertebral disc infection with Staphylococcus aureus evidenced the involvement of neutrophil response in apoptosis and pyroptosis of chondrocytes related to spondylodiscitis’ pathways [38]. Therefore, the interference of these pathways and the neutrophil dysfunctions during severe COVID-19 should be regarded in the future studies [39].

## 4. Conclusions

Spondylodiscitis with spinal abscess can be a rare but severe complication post-COVID-19 disease, due to dysbalanced immune response and following the respiratory infection, endothelial lesions, hypercoagulation and bacterial superinfection. Experience of a larger number of cases and additional studies are required to clarify the relationship of COVID-19 associated SD and reactive hemophagocytic lymphohistiocytosis. Secondary spinal infections and their complications should be considered in patients with spinal neuromuscular symptoms related to COVID-19, comprising neurologic and neurosurgical evaluation in diagnostic and monitoring COVID-19 protocols.

## Figures and Tables

**Figure 1 medicina-59-00616-f001:**
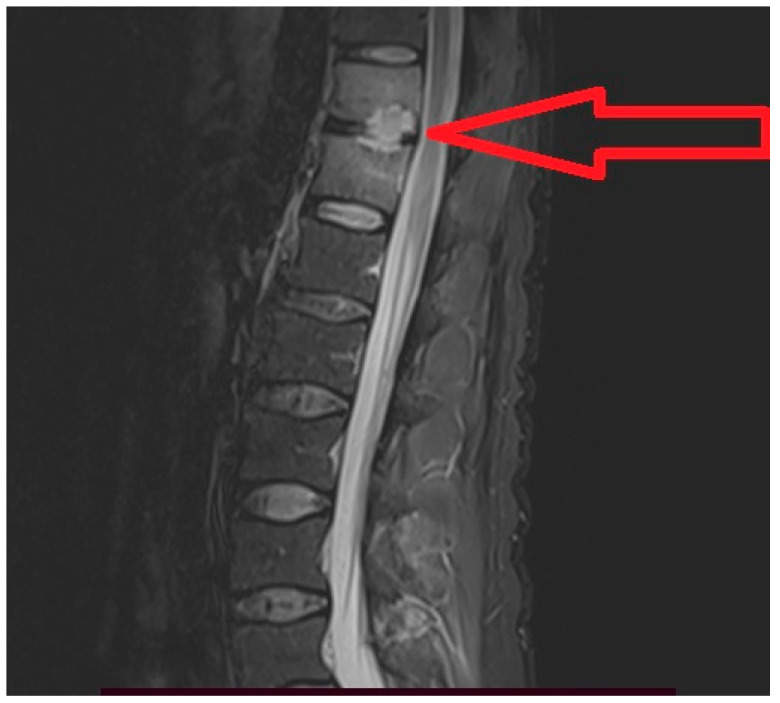
MRI sagittal section segment dorso-lumbar spine T2 TSE: Vertebral intraspongiosis bone edema T11 and T12 (hypersignal T2), disruption of the bone cortex adjacent to the T11 disc with disc damage.

**Figure 2 medicina-59-00616-f002:**
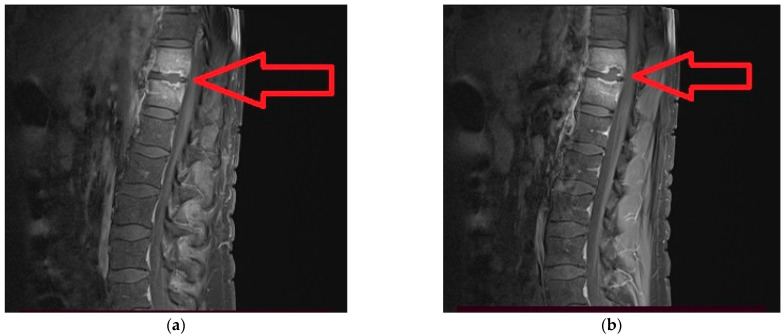
(**a**,**b**). MRI sagittal section segment dorso-lumbar spine T1 + C TSE FS: contrast enhancement at T11, T12 with intervertebral disc involvement. lesion process centred at the level of the intervertebral disc T11–T12 with extension at both adjacent vertebral plates causing osteolysis associating peripheral contrast enhancement and significant intraspongiosis oedema in both vertebral bodies concerned (T11 and T12).

**Figure 3 medicina-59-00616-f003:**
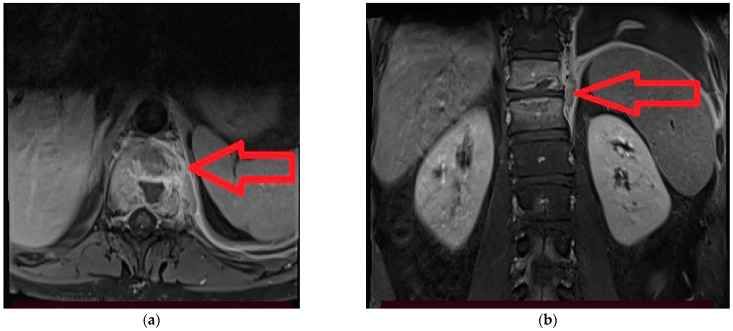
(**a**,**b**). MRI axial section dorsal spine segment T11 T1 + C TSE FS and fig 6. MRI coronal section dorso-lumbar spine segment T1 + C TSE FS: ‘’ring-like” contrast enhancement in the vertebral body with left paravertebral extension of the inflammatory process (paravertebral abscess).

**Table 1 medicina-59-00616-t001:** Characteristics of reported cases with Spondylodiscitis related to COVID-19 [27,28,29,30].

No	Gender	Age	Co-Morbidities	COVID-19 IST	Time Since COV	Fever	HC	Aetiology	Anatomic Site	Antibiotic	Surgery	Evolution	Dx Support	Reference
1	M	48	AHT Obesity	Tocilizumab HC	Conc. (ICU)	yes	neg	MSSA	Ab T1-T7	According ABG	Yes	Improve	MRI Ab + culture	Talamonti G, 2021 [27]
2	M	47	AHT Obesity	Tocilizumab Corticosteroid	Conc. (ICU)	no	poz	MSSA	Ab C4-C6	According ABG	Yes	Improve	MRI Ab + culture	Talamonti G, 2021 [27]
3	M	55	Lymphoma AHT, MI	Tocilizumab	Conc. (ICU)	no	neg	MSSA	Ab C5-T1	According ABG	Yes	Improve	MRI Ab + culture	Talamonti G, 2021 [27]
4	M	56	AHT, Venous Thrombosis	No	NA	no	neg	MSSA	Ab C1-2; Ab C7-T1	According ABG	Yes	Pulmonary infarction; deteriorate	MRI Ab + culture	Talamonti G, 2021 [27]
5	F	57	Anaemia	No	NA	no	poz	MSSA	Ab T12-L5	According ABG	Yes	Improve	MRI Ab + culture	Talamonti G, 2021 [27]
6	F	78	AHT, Obesity, Diabetes	No	NA	no	neg	MSSA	Ab T7-12	According ABG	Yes	Improve	MRI Ab + culture	Talamonti G, 2021 [27]
7	F	60	No	No	10 w	no	poz	MSSA	SD L4-5; Ab L3-2	Pip/Taz; Cefazolin; Cloxacillin *	Yes	Improve	MRI	Ramlee M, 2022 [30]
8	M	69	IC, Bone metastases, prostate cancer	No	12 w	no	neg	UN	SD L3-4-5	Ceftriaxone	Yes	Improve	MRI	Ramlee M, 2022 [30]
9	F	24	No	No	12 w	no	neg	TB	SD T2-7	HRZE	Yes	Improve	MRI, HP exam	Ramlee M, 2022 [30]
10	M	43	No	No	Conc (ICU)	yes	neg	UN	SD L3-4 Psoas abscess	NA	Yes	Improve	MRI	Erok B, 2021 [28]
11	M	71	No	HC	Conc (ICU)	yes	neg	UN	Multiple SD and Ab	Meronem + Vancomycin	Yes	MSOF Death	MRI	Naderi S, 2020 [29]

Legend: Ab = Abscess; ABG = Antibiogram; AHT = Arterial hypertension; Conc. = Concomitant; IC = ischaemic cardiopathy; HC = Hydroxychloroquine; HP = Histopathology; HRZE: Hydrazide, Rifampicin, Pyrazinamide, Ethambutol; ICU = Intensive Care Unit; IST = Immunosuppressive treatment; MI = myocardial infarction; MRI = Magnetic Resonance Imaging; NA = Not available; SD = Spondylodiscitis; TB = Tuberculosis; UN = unknown. * second line antibiotic, according susceptibility test.

## Data Availability

All data regarding the findings are available within the manuscript.

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
