# Peer review of "Post-COVID-19 Spondylodiscitis: A Case Study and Review of the Literature"

_medicina, 2023, doi:10.3390/medicina59030616_

Round 1

Reviewer 1 Report

Dear Authors,

congratulations on your valuable paper. Please, find here below some suggestions that, in my humble opinion, could further improve your work.

1. Please, consider to reword the title. As it appears now, it seems somewhat difficult to read. Maybe, it could be a good idea to reduce punctuation to a minimum.

2. Is it "coronaviral disease 2019" an official definition? I always read about "Coronavirus disease 2019" related to the acronym COVID19.

3. The introduction section presents a somewhat fragmented text, with a large amount of repetitions. I recommend a linguistic revision to improve the readability of the content.

4. I see that the main topic of your paper refers to SD rather than COVID19 but I think that the introduction is too wide with regard to SD and a little too concise with regard to COVID19. I would mention, for example, any recommended treatments for similar conditions. As well as the possible role of drugs approved for the treatment of COVID19 to date and their possible part in the occurrence of any adverse effects (which could also promote the occurrence of SD? Just wondering). I think the introductory part needs to be reworded a bit and improved.

5. Please, avoid repetitions. See, as an example, the exactly same beginnings of the two sentences at lines102 and 107 (but there are a few more cases).

6. I think that could be useful to add the timeframe of the patient's admission. It could be useful to better contestualize the treatments available and the different therapeutic options you took into consideration.

7. Why do you decide to use a 6 days course of remdesivir? Was it an off label treatment? Remdesivir was approved by the EMA for an initial 5-days course. I would recommend a clarification about this statement. Speaking of remdesivir again, please add references on the patient's liver and kidney function, which are necessary to begin to understand eligibility for the treatment.

8. Moreover, you did not mention if other COVID19 options were excluded and why. Please, add some comments about this.

9. Please, avoid the use of drugs' brand names (e.g. Clexane, to be replaced by enoxaparin).

10. Speaking of the Review of the Literature, you only mentioned a research in the Web of Science Core Collection. I would recommend an integratated research in Google Scholar and PubMed/Medline. Moreover, I would recommend to add a PRISMA flowchart in order to better represent graphically your search's results.

Author Response

REWIEVER 1

ANSWER TO COMMENTS AND SUGGESTIONS

  1. Please, consider to reword the title. As it appears now, it seems somewhat difficult to read. Maybe, it could be a good idea to reduce punctuation to a minimum.

R1: We have reworded the title:

Spondylodiscitis: A Rare Post-COVID-19 Spondylodiscitis:  Condition. A Case Study and Review of the Literature

  1. Is it "coronaviral disease 2019" an official definition? I always read about "Coronavirus disease 2019" related to the acronym COVID19.

R2: we revised.

  1. The introduction section presents a somewhat fragmented text, with a large amount of repetitions. I recommend a linguistic revision to improve the readability of the content.

R3: We revised the beginning of introduction section. The linguistic revision of a native English editor is required for the manuscript.

  1. I see that the main topic of your paper refers to SD rather than COVID-19 but I think that the introduction is too wide with regard to SD and a little too concise with regard to COVID19. I would mention, for example, any recommended treatments for similar conditions. As well as the possible role of drugs approved for the treatment of COVID19 to date and their possible part in the occurrence of any adverse effects (which could also promote the occurrence of SD? Just wondering). I think the introductory part needs to be reworded a bit and improved.

R4: We revised the introduction with supplementary data on COVID-19, in order to balance the information on COVID-19 and spondylodiscitis.

  1. Please, avoid repetitions. See, as an example, the exactly same beginnings of the two sentences at lines102 and 107 (but there are a few more cases).

R 5: We revised.

  1. I think that could be useful to add the timeframe of the patient's admission. It could be useful to better contestualize the treatments available and the different therapeutic options you took into consideration.

R6: We have added the notification of timeframe-April 2021, during the third pandemic wave.

  1. Why do you decide to use a 6 day course of remdesivir? Was it an off-label treatment? Remdesivir was approved by the EMA for an initial 5-days course. I would recommend a clarification about this statement. Speaking of remdesivir again, please add references on the patient's liver and kidney function, which are necessary to begin to understand eligibility for the treatment.

R7: Due to the severity of the pneumonia the patient received a supplementary dose of Remdesivir. A longer course than 5 days, but no more than 10 days, was allowed, according to the EMA summary of the product characteristics.

https://www.ema.europa.eu/en/documents/other/veklury-product-information-approved-chmp-25-june-2020-pending-endorsement-european-commission_en.pdf

The timetable of the liver and kidney function are presented in the supplementary material-Table A1. The kidney function was normal. The liver enzymes were increased x 2 NRV in the beginning of Remdesivir and was found over x 5 NRV in the 6th day, when we decided to stop Remdesivir.

  1. Moreover, you did not mention if other COVID19 options were excluded and why. Please, add some comments about this.

R8: When the patient was hospitalized for COVID-19, in April 2021, Favipiravir and Remdesivir were the available antiviral treatment options, approved by the national protocol of Minister of Health from Romania.

Romanian Ministery of Health. ORDIN Nr. 533/2021 din 22 aprilie 2021 privind modificarea anexei la Ordinul ministrului sănătăţii nr. 487/2020 pentru aprobarea protocolului de tratament al infecţiei cu virusul SARS-CoV-2.  MONITORUL OFICIAL NR. 434 din 23 aprilie 2021.

  1. Please, avoid the use of drugs' brand names (e.g. Clexane, to be replaced by enoxaparin).

R9: We revised.

  1. Speaking of the Review of the Literature, you only mentioned a research in the Web of Science Core Collection. I would recommend an integratated research in Google Scholar and PubMed/Medline. Moreover, I would recommend to add a PRISMA flowchart in order to better represent graphically your search's results.

We have completed the discussion section. The diagram flow is the supplementary Figure A3.

11 March 2023

Reviewer 2 Report

Dear Authors,

Thank you for giving me the opportunity to revise the paper entitled “Spondylodiscitis: A Rare Post-COVID-19 Condition. A Case 2 Study and Review of the Literature ”. The paper is well structured and succint and overall might give important contribution in new perspective in COVID19. Nevertheless, some critical issues should be addressed

Introduction: The introduction is quite confuse. The authors describe the possible diagnosis, cause and therapy of Spondylodiscitis in a very schematic manner, without link the possible link between COVID19 and Spondylodiscitis. Moreover, the authors should specify the aim of the study.

Case Report: Anamnestic patient characteristic should be reported

Discussion: well done

Best Regards

Author Response

REWIEVER 2

ANSWER TO COMMENTS AND SUGGESTIONS

Introduction: The introduction is quite confuse.

R: We revised the introduction in order to balance the information on COVID-19 and spondylodiscitis. English revision of a native English Editor is required to improve the clarity of the manuscript.

The authors describe the possible diagnosis, cause and therapy of Spondylodiscitis in a very schematic manner, without link the possible link between COVID19 and Spondylodiscitis.

R: The section “Discussions” are reviewing the other reported cases in the literature, the peculiarities of the case, with comments on the potential pathogenic mechanisms of spondylodiscitis associated to COVID-19.

We revised this section and have completed with the subsection Hypothesis of the linkage between COVID-19 and spondylodiscitis.

Moreover, the authors should specify the aim of the study.

R: We revised.

Case Report: Anamnestic patient characteristic should be reported

We have mentioned in the beginning of the case presentation that the patient had “a personal medical history of appendicectomy, duodenal ulcer and lumbar disc herniation”. No chronic medication or other chronic diseases were known.

11 March 2023

Round 2

Reviewer 1 Report

The authors addressed all the suggestions I made. Thank you very much.

Reviewer 2 Report

The authors have addressed all suggestio

Best Regards